# Identifying Plasma Biomarkers That Predict Patient-Reported Outcomes Following Treatment for Trapeziometacarpal Osteoarthritis Using Machine Learning

**DOI:** 10.3390/ijms26209856

**Published:** 2025-10-10

**Authors:** Mauro Maniglio, Moaath Saggaf, Nupur Purohit, Daniel Antflek, Jason S. Rockel, Mohit Kapoor, Heather L. Baltzer

**Affiliations:** 1Department of Hand Surgery, The Balgrist, University Clinic, 8008 Zürich, Switzerland; mauro.maniglio@balgrist.ch; 2Division of Plastic and Reconstructive Surgery, Department of Surgery, University of Toronto, Toronto, ON M5R 0A3, Canada; m.saggaf@mail.utoronto.ca; 3Department of Hand and Plastic Surgery, Toronto Western Hospital, UHN University of Toronto, Toronto, ON M5T 2S8, Canada; nupur.purohit@uhn.ca (N.P.); daniel.antflek@uhn.ca (D.A.); 4Osteoarthritis Research Program, Division of Orthopaedics, Schroeder Arthritis Institute, University Health Network, Toronto, ON M5T 0S8, Canada; jason.rockel@uhn.ca (J.S.R.); mohit.kapoor@uhn.ca (M.K.); 5Krembil Research Institute, University Health Network, Toronto, ON M5T 0S8, Canada; 6Department of Laboratory Medicine and Pathobiology, University of Toronto, Toronto, ON M5S 3K3, Canada

**Keywords:** plasma biomarkers, osteoarthritis, trapeziometacarpal joint, first carpometacarpal joint, CMC-1 osteoarthritis, TM osteoarthritis, trapeziometacarpal osteoarthritis, first carpometacarpal osteoarthritis

## Abstract

Osteoarthritis (OA) in the trapeziometacarpal joint (TM) is a prevalent form of hand OA, yet research on biomarkers specific to hand OA remains limited. This study aims to identify systemic plasma biomarkers at baseline in TM OA patients that are associated with patient-reported outcomes one year post-treatment. Blood samples and clinical data were collected prospectively from 143 TM OA patients undergoing conservative therapy, fat grafting, or surgery, with one-year follow-up. Supervised machine learning with Lasso regularization analyzed associations among 10 systemic biomarkers related to cartilage turnover, bone remodeling, pain, or lipid metabolism. Generalized estimating equation models evaluated baseline biomarker associations with one-year outcomes. Patients averaged 61 years, were mostly female (69%), and were primarily treated conservatively (47%). The machine learning model identified associations between outcomes and biomarkers, including PIIANP, Visfatin, adiponectin, and leptin. Adjusted analyses revealed baseline PIIANP associated with VAS, QuickDASH, and TASD improvements, while Visfatin correlated with VAS worsening. We could identify two different plasma markers that could predict the clinical outcome of TM OA treatment. Baseline PIIANP is associated with improvement, while Visfatin is associated with worsening in TM OA outcomes up to one year post-treatment. Further prospective studies are needed to confirm and solidify these findings.

## 1. Introduction

Osteoarthritis (OA) in the trapeziometacarpal joint (TM), also known as basilar thumb osteoarthritis, or first carpometacarpal osteoarthritis, is one of the most common locations of OA in the hand [1] and represents a major source of functional morbidity [2]. Its lifetime prevalence is around 10%, affecting women more frequently than men [3].

While age-related factors are major contributors [4], OA is now understood as a multifaceted joint degeneration process involving biomechanical [5], systemic [6], and local elements [1]. Repetitive thumb use, or radial subluxation, can increase joint stress, leading to cartilage wear [7] and subsequent bone remodeling [8].

After a clinical suspicion, the diagnosis of TM OA is based on plain X-rays. However, the radiographic OA prevalence and severity are often discordant with patient reported pain and functional assessments [9,10,11].

Based on radiological images, the TM OA prevalence is estimated to be 12–50%; however, a symptomatic TM OA affects 1–16% of individuals [12].

These discrepancies between radiological and clinical findings illustrate the need for objective markers linked to the disease severity. An understanding of such markers could offer the potential to achieve an earlier detection of TM OA, prediction of the progression and severity, and maybe targeting for new treatments [6,13,14].

It is known that the adipose tissue has metabolic activity and influences OA through an endocrine mechanism [1]. Therefore, obesity is also a well-established risk factor [15] of OA in the upper extremities.

With the clinical and therapeutic potential offered by identifying novel biomarkers [16], in addition to the lack of in-depth investigation into TM OA-specific systemic markers, we examined inflammatory cytokine markers [6], where we found that circulating cytokines are capable of distinguishing TM OA severity. Patients with TM OA and higher levels of IL-7 were associated with a decreased likelihood of needing surgical intervention. We also identified two distinct phenotypes, one inflammatory, based on the systemic cytokine signature of the TM OA cohort we had studied.

This prior work may target the inflammatory part of the disease controlled by the interleukins; however, this work did not examine other important biomarkers related to the lipid metabolism, cartilage turnover, bone remodeling, and pain.

Machine learning has become increasingly popular in handling large data sets to aid in identifying biomarkers [17], in part due to its capacity to take into consideration multiple factors and account for several confounders. Systemic biomarkers have already been described in other forms of OA [18] but studies of these biomarkers are lacking in TM OA.

Our study aimed to identify systemic biomarkers that predict patient-reported outcome measures (PROMs) indicative of pain and function at one year after treatment for TM OA, and test associations between identified systemic biomarkers and PROMs. In a sample of 343 subjects with primary symptomatic TM OA, we used machine learning with internal validation to identify plasma protein biomarkers indicative of pain and function. We identified a total of ten proteins known to be involved in lipid metabolism, cartilage turnover, bone remodeling, and pain and determined which out of these baseline plasma protein markers were associated with various TM OA PROMs. To our knowledge, this is the first study to apply a supervised machine learning framework to TM OA, integrating plasma biomarkers from multiple biological pathways with validated PROMs and linking them to both baseline status and longitudinal outcomes. This approach goes beyond previous cytokine-based studies by simultaneously considering markers of lipid metabolism, cartilage turnover, bone remodeling, and pain.

## 2. Results

A total of 143 patients with TM OA were included in the study, with 89 patients treated conservatively (splinting, corticosteroid injections, physiotherapy), 21 with autologous fat transplant, and 54 patients treated surgically with a trapeziectomy with ligament reconstruction and tendon interposition. Descriptive data on the baseline demographic, anthropometric, and clinical variables of our TM OA cohort can be found in Table 1. A total of 119 patients had OA in more than one joint in the body. At baseline a reciprocal correlation between some biomarkers was seen, which we have summarized in Figure 1.

We next used an internally validated, supervised machine learning model to identify associations between PROMs and concentrations of plasma biomarkers that varied consistently between baseline and final follow-up. At baseline (therapy initiation), Adiponectin had a negative association with TASD and Quick DASH, translating to increased adiponectin levels being associated with improved perceived functionality of the hand; however, this association was absent at the one-year follow-up. Leptin and CTX-1 (iC-telopeptide of type I collagen) were only associated with baseline clinical scores (TASD and Quick DASH) and not follow-up scores. Only PIIANP and visfatin had associations at baseline and one-year follow-up scores. Table 2 summarizes the associations from the machine learning model; all effect sizes greater than 0.01 were recorded.

Following the identification of candidate biomarkers associated with PROMs, the associations were subsequently formally tested analytically. In the adjusted analyses, baseline PIIANP was associated with improvements in the VAS (beta = −3.09, 95% CI: −6.07 to −0.11, *p* = 0.04), QuickDASH (beta = −3.99, 95% CI: −5.98 to −1.99, *p* < 0.0001), and TASD (beta = −2.42, 95% CI: −4.51 to −0.33, *p* = 0.02) longitudinally. Visfatin was associated with worsening in the VAS (beta = 3.04, 95% CI: 0.14 to 5.93, *p* = 0.04) after accounting for age, sex, BMI, and treatment method. Table 3 summarizes the analytical testing for candidate biomarkers.

## 3. Discussion

In OA research, the potential of prognostic biomarkers has captured significant attention. Different serum protein markers have been investigated [6]. According to the OA Biomarkers Network, a prognostic biomarker predicts the onset of OA in individuals without the disease and can also foresee the progression of OA in those already afflicted [19]. Unveiling novel biomarkers bears clinical and therapeutic promise. Additionally, biomarkers may better measure biological processes compared to radiological assessment. Despite this potential, there remains a lack of comprehensive exploration into TM OA-specific systemic biomarkers. Our study addresses this gap by being the first to systematically evaluate multiple classes of plasma biomarkers with advanced machine learning methods in a large TM OA cohort and by directly linking them to clinical outcomes after both surgical and nonsurgical treatment. This integrative approach highlights novel candidates such as PIIANP and Visfatin as prognostic indicators of pain and function in TM OA.

While our prior research focused on inflammation modulated by interleukins [6], certain essential biomarkers were left unexplored. Biomarkers tied to lipid metabolism, cartilage turnover, bone remodeling, and pain were not analyzed in this aforementioned study. Therefore, we examined ten candidates out of these for their potential of being a prognostic biomarker in TM OA.

In our internally validated supervised machine learning model, we identified an association between the baseline values of PIIANP and Visfatin with clinical outcomes at one-year follow-up. Higher baseline PIIANP was associated with an improvement in VAS, QuickDASH, and TASD symptoms of TM OA, whereas higher baseline Visfatin was associated with worsening of the VAS at one year following treatment. Our study showed a positive effect of higher PIIANP levels at baseline, with the PROMs of the disease one year after treatment, regardless of surgical, injection, or conservative treatment. PIIANP results from the production of Type II collagen, the most abundant cartilage protein, making up around 90% of the total collagen content [20]. Chondrocytes replace collagen with newly synthesized proteins to maintain cartilage in the joint, secreting procollagens into the extracellular matrix where specific proteinases remove extension propeptides, allowing incorporation of mature molecules into fibrils [20]. Among these, PIIANP, synthesized by chondrocytes in OA, is considered an anabolic marker of cartilage turnover, likely part of a repair attempt [20]. Thus, PIIANP levels in plasma may relate to repair responses that positively influence the patient’s outcome regardless of therapeutic modality.

In our study, higher Visfatin serum levels at baseline predicted higher pain levels one year after treatment. Visfatin, an adipokine secreted from adipose tissue, influences OA, in which serum levels are elevated [21]. Its association with inflammatory cytokines suggests involvement in the pro-inflammatory process of OA. In fact, Interleukin (IL)-1β and lipopolysaccharide (two important mediators of cartilage destruction) induce Visfatin expression, which in turn enhances the expressions of IL-6, tumor necrosis factor alpha (TNF-α), and matrix metalloproteinases (MMPs), implying a pro-degradative effect on cartilage in OA [22]. Thus, pro-inflammatory pathways may influence the pain perception negatively, and the described mechanism of cartilage destruction may expedite disease progression.

The strengths of the study include the fact that the cohort of patients was prospectively collected and the follow-up was standardized to minimize bias. In the machine learning model, we identified candidate biomarkers that were subsequently tested, accounting for important confounders, such as age, joint count, BMI, and sex. Building upon this foundation, we found associations between baseline plasma levels of PIIANP and Visfatin and the PROMs of TMOA during follow-up. Further studies are needed to bring these findings into clinical application.

However, our study is also not without limitations. For instance, the TM is small and may only modestly influence the plasma levels of biomarkers. Some patients in our cohort also had other larger joints with OA, which could have influenced the plasma levels of the proteins studied. To attempt to mitigate this, we adjusted for arthritic joint count as part of our analyses. Other factors, such as pharmacotherapy or rehabilitation, could have a potential influence on the plasma levels, which was not inspected in this study.

In addition, we had a theoretical risk of overfitting the machine learning model; however, we limited this risk with regularization and cross-validation.

In conclusion, we found an association of two plasma markers and TM OA progression of PROMs after surgical or non-surgical treatment. A higher baseline serum level of PIIANP was associated with favorable PROMs, whereas higher Visfatin levels were associated with an unfavorable evolution of pain at one-year follow-up. This exploration opens a promising avenue toward understanding relationships between circulating plasma proteins and TM OA and future precision medicine approaches in a shared decision-making model. High PIIANP and low Visfatin at baseline may potentially be used to predict a positive treatment outcome in TM OA.

## 4. Materials and Methods

### 4.1. Study Sample

All adult patients (N = 343) with a primary diagnosis of symptomatic OA in the TM were invited to prospectively donate clinical data and biospecimens to a TM OA biobank. Diagnosis was based on clinical diagnosis and radiographic signs of OA in the TM. Patients declining participation (N = 78) or those with an underlying inflammatory, crystalline, or post-traumatic arthritis condition (N = 7) or with insufficient data (N = 115) were excluded. Those with a minimal follow-up of 1 year were included in the study cohort (N = 143). Descriptive data on the demographics of our TM OA cohort (N = 143) can be found in Table 1.

Institutional Review Board approval was received at the University Health Network, Toronto, ON, Canada (REB# 17-5360).

At baseline (first consultation for TM symptoms), all participants donated plasma and completed a comprehensive electronic study questionnaire that documented age, sex, BMI, menopausal status, total afflicted joint count (total number of painful arthritic joints), and symptomatic status in the form of the Quick Disabilities of the Arm, Shoulder, and Hand (QuickDASH, which assesses disability by self-report of physical function and symptoms, from 0 no disability to 100 extremely disabled), the Trapeziometacarpal Arthrosis Symptoms and Disability Questionnaire (TASD which specifically assesses symptoms and disability at the TM, with a score of 0–100, with higher scores representing more severe levels of symptoms and/or disability) [23], and the Visual Analog Scale for pain (VAS, a horizontal scale of 10 cm labeled at each end by descriptors ‘no pain’ and ‘worst pain ever’ on which the patient reports their pain level). Radiographic images were scored according to the Eaton–Littler classification by consultant hand surgeons or hand surgery fellows. Baseline was defined as the first consultation in our clinic for TM OA symptoms and before conservative or surgical treatment. At 12 months post-treatment, participants completed identical follow-up questionnaires to allow the calculation of the difference from the baseline, representing any clinical changes.

### 4.2. Quantification of Protein Markers in Plasma

All patient plasma samples were quantified to determine the concentrations of biochemical protein markers known to be involved in the cascade of OA as part of cartilage turnover (N-propeptide of collagen IIA (PIIANP)), bone remodeling (Osteocalcin and C-telopeptide of type I collagen), fat metabolism (Adiponectin, Leptin, Adipsin, Visfatin), and pain (substance P, Beta-NGF, and brain-derived neurotrophic factor) using a multiplex assay (Bio-Rad, Mississauga, ON, Canada) and/or single-plex enzyme-linked immunosorbent assay (ELISA) kits (abcam, Waltham, MA, USA; Bio-Rad, Mississauga, ON, Canada; biotechne, Toronto, ON, Canada; MyBioSource Inc., San Diego, CA, USA; see Table A1). Anticoagulated whole blood samples were collected and immediately stored at 4 °C for between 4 and 24 h prior to being centrifuged at 4000 rpm for 10 min. Plasma was extracted, flash frozen in liquid nitrogen, and subsequently stored in liquid nitrogen tanks until use. Samples included in this study underwent single freeze/thaw cycles. Samples from the different groups and time points were randomly allocated to plates, and the experiment was conducted using de-identified samples run in duplicate on the Luminex 200 system (Luminex Corp., Austin, TX, USA) and analyzed using Luminex xPONENT 134 Software (v4.3, Luminex Corp., Austin, TX, USA).

### 4.3. Statistical Analyses

Continuous data were reported as means and standard deviations. Categorical data were reported as percentages. Systemic biomarkers were scaled to have a mean of zero and a standard deviation of one to assist in the interpretation of the results. Normality of continuous variables was assessed using the Shapiro–Wilk test and Q–Q plots.

We applied supervised machine learning using Lasso (least absolute shrinkage and selection operator) [24] regularization to identify associations among the 10 systemic biomarkers studied. Lasso was chosen because it performs both variable selection and regularization, thereby reducing the risk of overfitting and addressing multicollinearity among correlated biomarkers. To avoid information leakage, systemic biomarkers were standardized (mean = 0, SD = 1) within each cross-validation [25] fold. The regularization parameter (λ) was chosen using the minimum mean squared error criterion from repeated 10-fold cross-validation (50 repetitions). Prediction quality was evaluated based on cross-validated mean squared error and R^2^ (explained variance).

Outcome measures included the QuickDASH, VAS, and TASD questionnaires. Baseline values for systemic biomarkers were modeled to identify associations with baseline and one-year PROMs. Identified associations from the baseline and one-year outcomes were statistically tested in a longitudinal model. Associations that were not identified by the model were not tested to minimize the risk of type one error inflation with multiplicity. Joint count was explained by missingness based on the intervention and was imputed in a multiple imputation model under the assumption of missing at random. Generalized estimating equation [26] models were built for each outcome to assess associations between the identified baseline biomarkers and the one-year outcomes, accounting for age, sex, body mass index, joint count, and treatment. The final model handled missing PROMs data as missing completely at random. Significance level was set at *p* ≤ 0.05. R Statistics version 3.3.1 was used to run the statistical tests, and the machine learning protocols were run with Python (version 3.12), and the scikit-learn library was used for machine learning protocols.

## Figures and Tables

**Figure 1 ijms-26-09856-f001:**
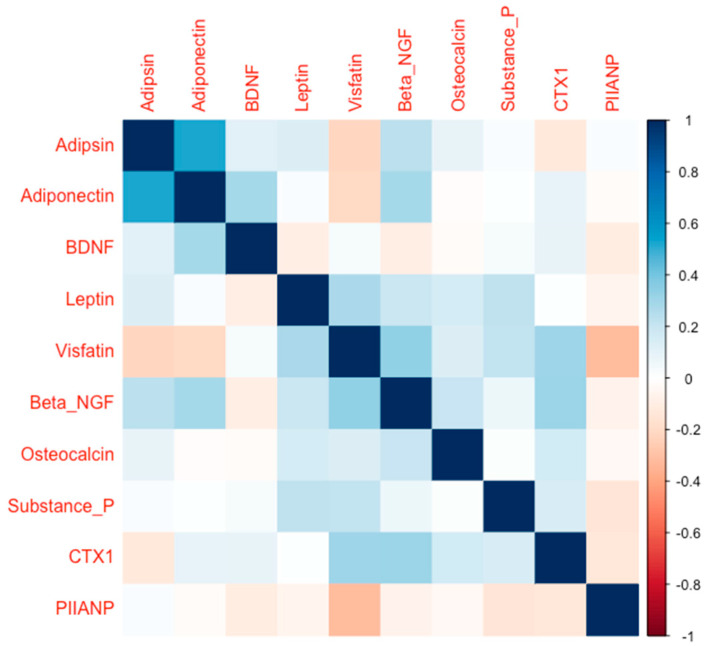
The correlation matrix of the studied biomarkers was measured at baseline. The blue color indicates a positive correlation, and the red color indicates a negative correlation. Darker colors indicate a stronger correlation compared to lighter colors.

**Table 1 ijms-26-09856-t001:** Baseline characteristics of the study participants.

Sex	N = 143
Female	99 (69%)
Male	44 (31%)
Age	N = 143
Mean (range)	61 (42–87)
BMI	N = 142
Mean (SD)	26.8 (5.6)
Eaton–Littler Grade	N = 138
1/2	44 (32%)
3	58 (42%)
4	36 (26%)
Scores at baseline	Mean (SD)
VAS Pain (N = 140)	61 (24)
QuickDASH Score (N = 142)	45 (19)
TASD Score (N = 143)	52 (19)
TASD Symptom Subscale (N = 143)	52 (19)
TASD Disability Subscale (N = 143)	53 (23)

SD = Standard deviation; VAS = Visual analog scale; DASH = Disabilities of the Arm, Shoulder, and Hand; TASD = Trapeziometacarpal Arthrosis Symptoms and Disability Questionnaire.

**Table 2 ijms-26-09856-t002:** Systemic biomarkers associated with baseline and one-year PROMs as identified by supervised machine learning and subsequent regression analyses.

Marker *	QuickDASH	VAS	TASD	TASD Symptom	TASD Disability
**Adipsin** Baseline One-year					
**Adiponectin** Baseline One-year	−1.49	−1.78	−3.63	−3.63	−2.71
**BDNF** Baseline One-year					
**Leptin** Baseline One-year	+0.01	+1.88	+0.58	+0.90	
**Visfatin** Baseline One-year	+0.77 −1.93	+3.32 −2.19	+1.26 −1.35	+0.81 −1.08	+1.26 −1.26
**Beta-NGF** Baseline One-year					
**Osteocalcin** Baseline One-year					
**Substance P** Baseline One-year					
**CTX-1** Baseline One-year		−0.91			
**PIIANP** Baseline One-year	−2.56 +1.21	+1.47	−0.37 +0.38	−0.23 +0.47	+1.16

* Empty cells indicate no association was identified from supervised machine learning. Biomarkers were first selected using Lasso regression with repeated 10-fold cross-validation. The regularization parameter (λ) was chosen using the minimum cross-validated mean squared error criterion. Feature stability was assessed by inclusion frequency across resamples. Effect estimates represent standardized regression coefficients (β) from generalized estimating equation (GEE) models, adjusted for age, sex, body mass index, joint count, and treatment. Estimates correspond to the change in PROM score per one standard deviation increase in biomarker level. Positive values indicate worse scores; negative values indicate better scores. Empty cells indicate no association identified by Lasso and therefore not tested. Abbreviations: BDNF, Brain-Derived Neurotrophic Factor; Beta-NGF, Beta-nerve growth factor; CTX-1, Type I Collagen Cross-Linked C-Telopeptide; QuickDASH, Quick Disabilities of the Arm, Shoulder, and Hand; PIIANP, N-propeptide of collagen IIA; TASD, Trapeziometacarpal Arthrosis Symptoms and Disability; VAS, Visual Analogue Scale.

**Table 3 ijms-26-09856-t003:** Results from the adjusted longitudinal analyses of baseline plasma biomarkers in relation to patient-reported outcomes at one-year follow-up. Estimates represent associations between biomarker concentrations at baseline and changes in outcome scores over time.

Outcome	Biomarker	Estimate * (95% CI)	*p*-Value
QuickDASH	Visfatin PIIANP Adiponectin Leptin	1.01 (−0.97 to 3.13) −3.99 (−5.98 to −1.99) −1.30 (−3.51 to 0.90) −0.02 (−2.31 to 2.26)	0.30 **<0.0001** 0.25 0.98
VAS	Visfatin PIIANP Adiponectin Leptin CTX-1	3.04 (0.14 to 5.93) −3.09 (−6.07 to −0.11) −2.76 (−5.92 to 0.39) 1.77 (−1.53 to 5.07) −1.23 (−4.13 to 1.67)	**0.04** **0.04** 0.09 0.29 0.41
TASD	Visfatin PIIANP Adiponectin Leptin	1.04 (−0.95 to 3.02) −2.42 (−4.51 to −0.33) −1.60 (−3.85 to 0.66) 0.64 (−1.64 to 2.93)	0.30 **0.02** 0.17 0.58
TASD Symptom Subscale	Visfatin PIIANP Adiponectin Leptin	0.69 (−1.16 to 2.54) −1.91 (−3.90 to 0.08) −1.59 (−3.79 to 0.63) 1.17 (−0.99 to 3.32)	0.47 0.06 0.16 0.29
TASD Disability Subscale	Visfatin PIIANP Adiponectin	1.52 (−0.92 to 3.96) −3.13 (−5.70 to −0.56) −1.62 (−4.26 to 1.01)	0.22 **0.02** 0.23

* Estimates represent standardized regression coefficients (β) derived from generalized estimating equation (GEE) models, adjusted for age, sex, body mass index, joint count, and treatment type. Biomarker concentrations were standardized (mean = 0, SD = 1) prior to modeling. Thus, the coefficients correspond to the expected change in PROM score (QuickDASH, VAS, TASD, or subscales) per one standard deviation increase in biomarker level. Negative values indicate improved outcomes (lower pain or disability), while positive values indicate worse outcomes. A *p*-value of ≤0.05 is **bolded** to indicate significance.

## Data Availability

The data presented in this study are available on request from the corresponding author due to privacy protections placed on the study data obtained from human participants.

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
