# Peer review of "Identifying Plasma Biomarkers That Predict Patient-Reported Outcomes Following Treatment for Trapeziometacarpal Osteoarthritis Using Machine Learning"

_ijms, 2025, doi:10.3390/ijms26209856_

Round 1
Reviewer 1 Report
Comments and Suggestions for Authors
The article presents valuable and up-to-date research on the identification of plasma biomarkers associated with treatment outcomes in patients with trapeziometacarpal osteoarthritis (TM OA). The authors applied machine learning methods to determine potential prognostic markers, which gives the study a modern and interdisciplinary character. The results, indicating the role of PIIANP and Visfatin as factors associated with clinical improvement and deterioration, respectively, are promising and may provide a basis for further translational research. The manuscript is clear, well-structured, and the applied methods (biomarkers, machine learning, statistical analysis) are appropriate for the stated objectives. The text makes an important contribution to the advancement of knowledge on biomarkers in thumb osteoarthritis, a condition that has so far been relatively rarely investigated. Before further processing, the paper requires only minor revisions and additions. Detailed comments are provided below.
Minor comments:
Expanding the introduction with a more thorough discussion of osteoarthritis would improve the section by emphasizing both its clinical significance and its societal burden. Osteoarthritis is a complex, multifactorial disease influenced by factors such as occupational stress, sports activity, previous injuries, obesity, and sex-related differences. A concise overview of these risk factors, combined with the clinical context of knee osteoarthritis—particularly the importance of early diagnosis and the limitations of subjective clinical evaluation—would provide a solid foundation for the study. To strengthen this part, the following references are suggested: https://doi.org/10.1007/174_2024_516 ; https://doi.org/10.3390/app15126896 ; https://doi.org/10.3390/healthcare12161648 ; https://doi.org/10.1038/s41584-025-01223-y;
The authors mentioned the small impact of a single joint on biomarker levels and the risk of model overfitting, but the issue of patient heterogeneity (including the presence of OA changes in other joints) and the potential impact of concomitant treatment (e.g., pharmacotherapy, rehabilitation) should be discussed in greater detail. The limitations section should be expanded to discuss factors that may interfere with the interpretation of plasma biomarkers.
The paper documents statistical relationships well, but less space is devoted to their practical application (e.g., the possibility of using markers in the process of qualifying for surgical treatment or monitoring the effects of therapy). In my opinion, it would be worthwhile to supplement the discussion with potential clinical scenarios for the use of PIIANP and Visfatin, as well as a comparison with biomarkers used in OA of other locations.
The authors mention Lasso and cross-validation, but the description remains rather brief. For example, there is no detailed justification for the choice of model, description of regularization parameters, or prediction quality measures. Please add more details in the “Materials and Methods” section to increase the transparency of the analysis and the possibility of its replication.
There are minor editorial inaccuracies in places (e.g., typos, repetitions in section titles, double “Correspondence”).
Although the article refers to current sources, the bibliography could be supplemented with the latest works on OA biomarkers (2020–2025) to better contextualize the study in the context of current trends. I recommend adding: DOI: 10.3390/pr11041014; DOI: 10.3390/app10238312; https://doi.org/10.1016/j.berh.2023.101852; https://doi.org/10.3390/life12111799; https://doi.org/10.1016/j.joca.2022.09.005
Author Response
Comments and Suggestions for Authors
The article presents valuable and up-to-date research on the identification of plasma biomarkers associated with treatment outcomes in patients with trapeziometacarpal osteoarthritis (TM OA). The authors applied machine learning methods to determine potential prognostic markers, which gives the study a modern and interdisciplinary character. The results, indicating the role of PIIANP and Visfatin as factors associated with clinical improvement and deterioration, respectively, are promising and may provide a basis for further translational research. The manuscript is clear, well-structured, and the applied methods (biomarkers, machine learning, statistical analysis) are appropriate for the stated objectives. The text makes an important contribution to the advancement of knowledge on biomarkers in thumb osteoarthritis, a condition that has so far been relatively rarely investigated. Before further processing, the paper requires only minor revisions and additions. Detailed comments are provided below.
Response: We thank the reviewer for this positive feedback
Minor comments:
Comment 1: Expanding the introduction with a more thorough discussion of osteoarthritis would improve the section by emphasizing both its clinical significance and its societal burden. Osteoarthritis is a complex, multifactorial disease influenced by factors such as occupational stress, sports activity, previous injuries, obesity, and sex-related differences. A concise overview of these risk factors, combined with the clinical context of knee osteoarthritis—particularly the importance of early diagnosis and the limitations of subjective clinical evaluation—would provide a solid foundation for the study. To strengthen this part, the following references are suggested: https://doi.org/10.1007/174_2024_516 ; https://doi.org/10.3390/app15126896 ; https://doi.org/10.3390/healthcare12161648 ; https://doi.org/10.1038/s41584-025-01223-y;
Response 1: We expanded the introduction including some new references.
Adding: “After a clinical suspicion the diagnosis of TM OA is based on plain x-rays. However, the radiographic OA prevalence and severity is often discordant with patient reported pain and functional assessments9-11.
Based on radiological images the TM OA prevalence is estimated to be of 12-50%, however a symptomatic TM OA affects 1-16% of individuals12.
These discrepancies between radiological and clinical findings illustrate the need for objective markers linked to the disease severity. An understanding of such markers could offer the potential to achieve an earlier detection of TM OA, prediction of the progression and severity and maybe targeting for new treatments6; 13; 14.”
And:
“With the clinical and therapeutic potential offered by identifying novel biomarkers, in addition to the lack of in-depth investigation into TM OA-specific systemic markers, we examined inflammatory cytokine markers6, where we found that circulating cytokines are capable of distinguishing TM OA severity. Patients with TM OA and higher levels IL-7 were associated with a decreased likelihood of needing surgical intervention. We also identified two distinct phenotypes, one inflammatory, based on the systemic cytokine signature of the TM OA cohort we had studied.
This prior work may target the inflammatory part of the disease controlled by the interleukins; however, this work did not examine other important biomarkers related to the lipid metabolism, cartilage turnover, bone remodeling and pain. “
Comment 2: The authors mentioned the small impact of a single joint on biomarker levels and the risk of model overfitting, but the issue of patient heterogeneity (including the presence of OA changes in other joints) and the potential impact of concomitant treatment (e.g., pharmacotherapy, rehabilitation) should be discussed in greater detail. The limitations section should be expanded to discuss factors that may interfere with the interpretation of plasma biomarkers.
Response 2: These are valuable points. We took this in consideration and that’s. So, we added the word arthritic to further clarify this point “To attempt to mitigate this, we adjusted for arthritic joint count as part of our analyses.” And we added the following limitation: “Also other factor as pharmacotherapy or rehabilitation could have a potential influence on the plasma levels, which was not inspected in this study.”
Comment 3: The paper documents statistical relationships well, but less space is devoted to their practical application (e.g., the possibility of using markers in the process of qualifying for surgical treatment or monitoring the effects of therapy). In my opinion, it would be worthwhile to supplement the discussion with potential clinical scenarios for the use of PIIANP and Visfatin, as well as a comparison with biomarkers used in OA of other locations.
Response 3: Because our study only lays the foundaments to such diagnostic options, we didn’t expand these scenarios too much. But following the recommendation of the Reviewer we added following sentence to the conclusion: “High PIIANP and low Visfatin at baseline can be potentially used to predict a positive treatment outcome in TM OA.”
Comment 4: The authors mention Lasso and cross-validation, but the description remains rather brief. For example, there is no detailed justification for the choice of model, description of regularization parameters, or prediction quality measures. Please add more details in the “Materials and Methods” section to increase the transparency of the analysis and the possibility of its replication.
Response 4: We thank the reviewer for this valuable comment. We agree that the description of our machine learning approach was too brief and have now expanded the “Materials and Methods” section to include a justification for using Lasso regularization, details on the choice of regularization parameters, and information on prediction quality measures. In short, Lasso was selected because it performs variable selection in settings with potential multicollinearity among predictors, which was expected in our dataset of systemic biomarkers. The optimal regularization parameter (λ) was determined by repeated 10-fold cross-validation, minimizing the mean squared error. Model performance was assessed using cross-validated mean squared error and the proportion of variance explained (R²). These details have been added to the revised manuscript to increase transparency and replicability:
“We applied supervised machine learning using Lasso (least absolute shrinkage and selection operator) regularization to identify associations among the 10 systemic biomarkers studied. Lasso was chosen because it performs both variable selection and regularization, thereby reducing the risk of overfitting and addressing multicollinearity among correlated biomarkers. The regularization parameter (λ) was selected through repeated 10-fold cross-validation, minimizing the cross-validated mean squared error. Prediction quality was evaluated based on cross-validated mean squared error and R² (explained variance).”
Comment 5: There are minor editorial inaccuracies in places (e.g., typos, repetitions in section titles, double “Correspondence”).
Response 5: We reviewed the whole manuscript and corrected all typos etc. we found
Comment 6: Although the article refers to current sources, the bibliography could be supplemented with the latest works on OA biomarkers (2020–2025) to better contextualize the study in the context of current trends. I recommend adding: DOI: 10.3390/pr11041014; DOI: 10.3390/app10238312; https://doi.org/10.1016/j.berh.2023.101852; https://doi.org/10.3390/life12111799; https://doi.org/10.1016/j.joca.2022.09.005
Response 6: We added some proposed citations.
Reviewer 2 Report
Comments and Suggestions for Authors
- Add a clear flow chart (recruitment → exclusions → inclusion) and list each exclusion reason with corresponding numbers; report baseline comparability across treatment subgroups.
- Handle age, sex, BMI, Eaton grade, and treatment modality consistently within the model; consider stratifying by treatment modality or including interaction terms to mitigate treatment-selection bias.
- Specify that standardization is performed within the CV folds, clarify the λ selection criterion (min vs. 1-SE), indicate whether repeated CV/bootstrapping was used, and report feature stability (e.g., inclusion frequency).
- "Machine learning has become in creasingly popular in handling big data sets to aid in identifying biomarkers (page 2, lines 52-53)", Please provide references for this statement. For example, Aborode AT, Emmanuel OA, Onifade IA, Olotu E, Otorkpa OJ, Mehmood O, Abdulai SI, Jamiu A, Osinuga A, Oko CI, Fakorede S, Mangdow M, Babatunde O, Olapade Z, Victoria AG, Salami A, Usman IA, Agboli VI, Adesola RO. The role of machine learning in discovering biomarkers and predicting treatment strategies for neurodegenerative diseases: A narrative review. NeuroMarkers. 2025;1. https://doi.org/10.1016/j.neumar.2024.100034;Gao Y, Mu J, Liu K, Wang M. Integrating molecular fingerprints with machine learning for accurate neurotoxicity prediction: an observational study. Adv Technol Neurosci. 2025;2(3):109-115.https://doi.org/10.4103/ATN.ATN-D-24-00034
Author Response
Comments and Suggestions for Authors
Comment 1: Add a clear flow chart (recruitment → exclusions → inclusion) and list each exclusion reason with corresponding numbers; report baseline comparability across treatment subgroups.
Response 1: The reasons of exclusion are already outlined in brackets; we don’t believe that more details would be useful for readership. But we added a table outlining the demographics of the cohort, that gives more information about the different subgroups.
Comment 2: Handle age, sex, BMI, Eaton grade, and treatment modality consistently within the model; consider stratifying by treatment modality or including interaction terms to mitigate treatment-selection bias.
Response 2: Our model already had accounted for “for age, sex, body mass index, joint count and treatment” as mentioned in line 269
Comment 3: Specify that standardization is performed within the CV folds, clarify the λ selection criterion (min vs. 1-SE), indicate whether repeated CV/bootstrapping was used, and report feature stability (e.g., inclusion frequency).
Response 3: We thank the reviewer for the helpful suggestions. We now provide additional details regarding our Lasso procedure. Standardization of systemic biomarkers was performed within each cross-validation fold to avoid information leakage. The regularization parameter λ was selected according to the minimum mean squared error criterion from the cross-validation. We applied repeated 10-fold cross-validation (50 repetitions) to improve robustness. These clarifications have been added to the “Materials and Methods” section to improve transparency and reproducibility.
Comment 4: "Machine learning has become in creasingly popular in handling big data sets to aid in identifying biomarkers (page 2, lines 52-53)", Please provide references for this statement. For example, Aborode AT, Emmanuel OA, Onifade IA, Olotu E, Otorkpa OJ, Mehmood O, Abdulai SI, Jamiu A, Osinuga A, Oko CI, Fakorede S, Mangdow M, Babatunde O, Olapade Z, Victoria AG, Salami A, Usman IA, Agboli VI, Adesola RO. The role of machine learning in discovering biomarkers and predicting treatment strategies for neurodegenerative diseases: A narrative review. NeuroMarkers. 2025;1. https://doi.org/10.1016/j.neumar.2024.100034;Gao Y, Mu J, Liu K, Wang M. Integrating molecular fingerprints with machine learning for accurate neurotoxicity prediction: an observational study. Adv Technol Neurosci. 2025;2(3):109-115.https://doi.org/10.4103/ATN.ATN-D-24-00034
Response 4: Thanks for this important remark. We added the first suggested reference.
Reviewer 3 Report
Comments and Suggestions for Authors
Trapeziometacarpal osteoarthritis (TMC OA) is a common condition affecting the joint at the base of the thumb, leading to pain and functional impairment. Identifying plasma biomarkers to predict treatment outcomes for TMC OA can enhance personalized treatment strategies. Machine learning (ML) can play a crucial role in analyzing complex data sets and identifying patterns that may not be evident through traditional statistical methods. This approach could lead to more tailored therapies, improved patient management, and enhanced understanding of the disease's underlying mechanisms. Therefore, the study of dr. Maniglio et al. is very important.
Comments
- All the typos and all the repeats should be corrected. All the abbreviations should be disclosed on first use.
- Plasma Biomarkers for TMC OA involve inflammatory markers such as CRP, IL-1β, Il-6, TNFα etc. It is not clear why proinflammatory markers were not examined in the study? This should be clarified.
- Title versus Abstract: There is a discrepancy: the Title states that biomarkers had been identified while the Abstract suggests that the obtained associations could aid in predicting PROM. This should be corrected.
- Line 62: It is not clear how a total 10 proteins were identified? This should be clarified.
- Line 71; Fig 1: Why these biomarkers were chosen? This should be clarified.
- Table 3 requires a footnote explaining I detail, how the “Estimates*” were obtained. This should be clarified.
- Table 3 caption should be clarified too.
- Line 157: The number of patients which had OA of other joints should be indicated at the beginning of the Results section.
- Conclusion: It is not clear whether high or low baseline PIIANP or Visfatin were associated with favorable versus unfavorable PROMs, respectively. This should be clarified.
- Methods section should be supplemented with references on all the methods used in the study. All the protocols used should be described in detail.
- Section 4.3: Normality test results should be also presented.
- The novelty of the study should be underlined.
Author Response
Comments and Suggestions for Authors
Trapeziometacarpal osteoarthritis (TMC OA) is a common condition affecting the joint at the base of the thumb, leading to pain and functional impairment. Identifying plasma biomarkers to predict treatment outcomes for TMC OA can enhance personalized treatment strategies. Machine learning (ML) can play a crucial role in analyzing complex data sets and identifying patterns that may not be evident through traditional statistical methods. This approach could lead to more tailored therapies, improved patient management, and enhanced understanding of the disease's underlying mechanisms. Therefore, the study of dr. Maniglio et al. is very important.
Response: We thank you for the positive feedback.
Comments
Comment 1: All the typos and all the repeats should be corrected. All the abbreviations should be disclosed on first use.
Response 1: We corrected typos and mentioned the missing abbreviation at the first time of use.
Comment 2: Plasma Biomarkers for TMC OA involve inflammatory markers such as CRP, IL-1β, Il-6, TNFα etc. It is not clear why proinflammatory markers were not examined in the study? This should be clarified.
Response 2: In a prior study we examined these factors. We added this information in the introduction:
“With the clinical and therapeutic potential offered by identifying novel biomarkers, in addition to the lack of in-depth investigation into TM OA-specific systemic markers, we examined inflammatory cytokine markers6, where we found that circulating cytokines are capable of distinguishing TM OA severity. Patients with TM OA and higher levels IL-7 were associated with a decreased likelihood of needing surgical intervention. We also identified two distinct phenotypes, one inflammatory, based on the systemic cytokine signature of the TM OA cohort we had studied.”
Comment 3: Title versus Abstract: There is a discrepancy: the Title states that biomarkers had been identified while the Abstract suggests that the obtained associations could aid in predicting PROM. This should be corrected.
Resposne 3: The reviewer is right. We resolved the discrepancy changing the abstract.
Comment 4: Line 62: It is not clear how a total 10 proteins were identified? This should be clarified.
Response 4: The proteins were identified in other studies to be involved in OA (mostly in bigger joints). We added following sentence to clarify: “known to be involved in the cascade of OA”
Comment 5: [Line 71; Fig 1: Why these biomarkers were chosen? This should be clarified.
Response 5: We added it in the main text. See above
Comment 6: Table 3 requires a footnote explaining I detail, how the “Estimates*” were obtained. This should be clarified.
Response 6: We thank the reviewer for this comment and agree that further clarification is required. We have now revised the footnote of Table 3 to specify how the estimates were obtained. Specifically, the reported effect estimates are standardized regression coefficients derived from generalized estimating equation (GEE) models, adjusted for age, sex, BMI, joint count, and treatment. These coefficients correspond to the change in patient-reported outcome measure (PROM) score per one standard deviation increase in biomarker level. The revised table footnote clarifies these details.
Comment 6: Table 3 caption should be clarified too.
Response 6: We thank the reviewer for pointing this out. We have now revised the caption of Table 3 to provide a clearer description of the content and the derivation of the estimates.
Comment 7: Line 157: The number of patients which had OA of other joints should be indicated at the beginning of the Results section.
Response 7: We added this information.
Comment 8: Conclusion: It is not clear whether high or low baseline PIIANP or Visfatin were associated with favorable versus unfavorable PROMs, respectively. This should be clarified.
Response 8: We thank the reviewer to pointing out this important inprecision. We change the passage to make it clearer: “A higher baseline serum level of PIIANP was associated with a favorable PROMs, whereas higher Visfatin levels were associated with an unfavorable evolution of pain at one year follow-up.”
Comment 9: Methods section should be supplemented with references on all the methods used in the study. All the protocols used should be described in detail.
Response 9: We thank the reviewer for this important comment. We have now supplemented the Methods section with references for analytical approaches. This increases the transparency and reproducibility of our work.
Comment 10: Section 4.3: Normality test results should be also presented.
Response 10: We thank the reviewer for this comment. We now report the results of normality testing in the Statistical Analyses section. Normality of continuous variables was assessed using the Shapiro–Wilk test and inspection of Q–Q plots.
Comment 11: The novelty of the study should be underlined.
Response 11: We thank the reviewer for this comment. We have revised the Introduction and Discussion to explicitly underline the novelty of our study:
We added following paragraph in the introduction:
“To our knowledge, this is the first study to apply a supervised machine learning framework to TM OA, integrating plasma biomarkers from multiple biological pathways with validated PROMs and linking them to both baseline status and longitudinal outcomes. This approach goes beyond previous cytokine-based studies by simultaneously considering markers of lipid metabolism, cartilage turnover, bone remodeling, and pain.”
And following to the discussion:
“Our study addresses this gap by being the first to systematically evaluate multiple classes of plasma biomarkers with advanced machine learning methods in a large TM OA cohort, and by directly linking them to clinical outcomes after both surgical and nonsurgical treatment. This integrative approach highlights novel candidates such as PIIANP and visfatin as prognostic indicators of pain and function in TM OA.”